# OR-LLM-Agent: Automating Modeling and Solving of Operations Research Optimization Problems with Reasoning LLM

## Abstract

With the rise of artificial intelligence (AI), applying large language models (LLMs) to mathematical problem-solving has attracted increasing attention. Most existing approaches attempt to improve Operations Research (OR) optimization problem-solving through prompt engineering or fine-tuning strategies for LLMs. However, these methods are fundamentally constrained by the limited capabilities of non-reasoning LLMs. To overcome these limitations, we propose OR-LLM-Agent, an AI agent framework built on reasoning LLMs for automated OR problem solving. The framework decomposes the task into three sequential stages: mathematical modeling, code generation, and debugging. Each task is handled by a dedicated sub-agent, which enables more targeted reasoning. We also construct BWOR, an OR dataset for evaluating LLM performance on OR tasks. Our analysis shows that in the benchmarks NL4OPT, MAMO, and IndustryOR, reasoning LLMs sometimes underperform their non-reasoning counterparts within the same model family. In contrast, BWOR provides a more consistent and discriminative assessment of model capabilities. Experimental results demonstrate that OR-LLM-Agent utilizing DeepSeek-R1 in its framework outperforms advanced methods, including GPT-o3, Gemini 2.5 Pro, DeepSeek-R1, and ORLM, by at least 7% in accuracy. These results demonstrate the effectiveness of task decomposition for OR problem solving.

## 1 Introduction

Operations Research (OR) plays a vital role in addressing complex decision-making challenges faced by businesses and industries Saban & Weintraub (2021); DeCroix et al. (2021). By formulating mathematical models and applying optimization algorithms, OR enhances efficiency and maximizes economic benefits across various domains. However, translating real-world problems into solvable mathematical models remains a significant challenge, as they are typically described in natural language rather than structured mathematical form. Bridging this gap requires domain expertise to systematically set up math models, and solving such models requires programming skills for coding, solver configuration, and debugging. These demands create barriers for non-expert users and hinder the practical deployment of OR techniques.

The rapid advancement of artificial intelligence (AI), particularly the emergence of large language models (LLMs), has introduced new opportunities to address these challenges. LLMs have shown strong capabilities in natural language understanding (Ouyang et al., 2022). Through training on large-scale textual data, they have also acquired substantial knowledge in domains such as mathematics and programming. These LLMs, such as GPT-4 (Achiam et al., 2023) and DeepSeek-V3 (Liu et al., 2024), not only comprehend human instructions but have also shown the ability to solve simple logical tasks, such as mathematical problems (Abdin et al., 2024; Li et al., 2025) and programming tasks (Anysphere, 2025). Furthermore, the development of reasoning LLMs, such as GPT-o1 (OpenAI, 2024b) and DeepSeek-R1 (Guo et al., 2025), has enhanced their capacity for systematic and structured reasoning, enabling more rigorous problem-solving in both programming and mathematical domains. Compared to non-reasoning LLMs like GPT-4, these reasoning LLMs exhibit substantially stronger mathematical and coding capabilities (Guo et al., 2025).

With these advances, OR tasks that traditionally required expert involvement are now increasingly being automated by AI systems. However, existing research on applying LLMs to OR still faces several challenges. First, recent studies (Huang et al., 2025; Xiao et al., 2023; AhmadiTeshnizi et al., 2024) attempt to enhance OR performance through fine-tuning or complex prompt engineering, but primarily rely on non-reasoning LLMs, which fundamentally limit their reasoning capabilities. Second, existing benchmark datasets for OR tasks are still limited in number and scope, calling for more diverse and reliable benchmarks.

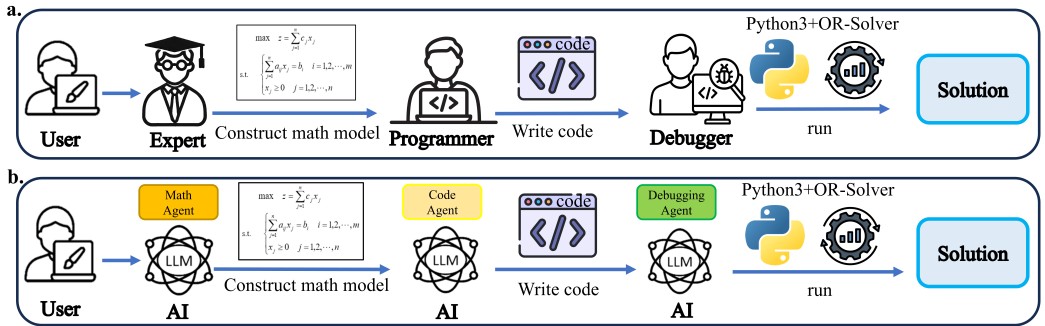

Figure 1: Existing expert solution (a) vs. proposed OR-LLM-Agent framework (b).

To address these challenges, we propose OR-LLM-Agent, a fully automated AI agent framework that solves OR problems using reasoning LLMs. Inspired by human expert workflows, OR-LLM-Agent decomposes OR tasks into three stages: mathematical modeling, code generation, and debugging, as illustrated in Figure 1. Each stage is handled by a dedicated sub-agent powered by an off-the-shelf reasoning LLM, without any retraining. This design enhances consistency and solution quality. We also construct BWOR, an OR dataset for more accurate evaluation of model performance. Our main contributions are as follows:

- We propose OR-LLM-Agent, an AI agent framework for automatic modeling, coding, debugging, and solving of OR problems based on reasoning LLMs, without requiring any additional fine-tuning or retraining.

- We demonstrate that decomposing OR tasks into three subtasks and assigning each to a dedicated sub-agent built on a reasoning LLM leads to better performance, without relying on complex prompt engineering.

- We construct BWOR, a new benchmark dataset for evaluating LLMs on OR tasks. Our analysis reveals that, in the benchmarks NL4OPT, MAMO, and IndustryOR, reasoning LLMs often underperform their non-reasoning counterparts within the same model family. In contrast, BWOR more effectively distinguishes differences in model performance.

- In the experiments, we evaluate various reasoning LLMs and state-of-the-art (SOTA) methods across five OR datasets. The results show that OR-LLM-Agent utilizing DeepSeek-R1 in its framework outperforms other methods such as GPT-o3 and ORLM by at least 7%.

## 2 RELATED WORKS

**LLM-Based Code Generation and Debugging:** Large Language Models (LLMs) have shown strong abilities in translating natural language into executable code. OpenAI's Codex (Chen et al., 2021) achieved impressive single-pass accuracy on HumanEval. AlphaCode (Li et al., 2022) advanced this with large-scale sampling and test-based filtering, reaching near-human performance in programming contests. AutoGen (Wu et al., 2023) introduced a multi-agent framework, where code agents iteratively collaborate with testing and debugging agents. OpenCodeInterpreter (Zheng et al., 2024) unified code generation, execution, and optimization, reaching 91.6% accuracy on HumanEval using synthesized feedback. However, these systems are not tailored for Operations Research (OR), limiting their ability to model domain-specific tasks or generate solver-compatible code.

**LLM-Based methods for OR:** Recent efforts apply LLMs to OR through agent collaboration, synthetic data, and interactive tools. Chain-of-Experts (Xiao et al., 2023) uses OR-informed agents to solve linear programs, while ORLM (Huang et al., 2025) trains LLMs on synthetic pipelines, achieving strong performance on NL4OPT (Ramamonjison et al., 2022), MAMO (Huang et al., 2024), and IndustryOR (Huang et al., 2025). OptiMUS (AhmadiTeshnizi et al., 2024) extracts (MI)LPs from long text via a modular agent, with 20–30% accuracy gains. OptiGuide (Li et al., 2023) enables "what-if" analysis through LLM-integrated solvers, and OptiChat (Ramamonjison et al., 2022) supports GPT-4-based debugging of infeasible models. Yet, most of these systems rely on non-reasoning LLMs, requiring manual chain-of-thought prompts or fine-tuning to overcome reasoning gaps—still limiting overall performance.

## 3 PRELIMINARIES

This section provides the foundational concepts relevant to LLMs. We introduce the features of reasoning LLMs and the prompt templates.

**Reasoning LLMs**: Before generating an answer, a reasoning LLM performs an explicit reasoning process, enclosed within the special tags $< think >$ and $< /think >$. During this stage, the model systematically analyzes the problem and verifies its reasoning through self-reflection (Guo et al., 2025). Once the reasoning is complete, the model synthesizes its thoughts and produces a final answer. In contrast, non-reasoning LLMs lack a dedicated reasoning phase. They generate responses in a token-by-token manner without structured deliberation.

**Prompt templates**: LLMs are typically prompted using one of 3 templates (OpenAI, 2024a).

- System prompt: The system prompt is used to set the global context and behavioral guidelines for the entire conversation or task.
- User prompt: The user prompt directly conveys actual questions or task requirements and serves as input to the model's response generation.
- Assistant prompt: The assistant prompt is generated by the LLMs to display its reasoning process and the answer.

## 4 METHODOLOGY

OR-LLM-Agent imitates human experts by dividing OR problem-solving into three stages: modeling, code generation, and execution/debugging—handled by the Math, Code, and Debugging Agents, respectively. The Math Agent translates natural language into a mathematical model, the Code Agent converts it into solver code, and the Debugging Agent executes and repairs the code as needed. The workflow is illustrated in Figure 2.

### 4.1 DEFINITIONS

**Math Agent:** An LLM that converts OR tasks expressed in natural language into mathematical models.

$$f_{\mathrm{math}} : \mathcal{T} \to \mathcal{M}, \quad T \mapsto M, \tag{1}$$

where $\mathcal{T}$ denotes the space of natural language OR tasks, and $T \in \mathcal{T}$ is a specific task. $\mathcal{M}$ is the space of mathematical models, and $M \in \mathcal{M}$ is a specific model for task $T$.

**Code Agent:** An LLM that converts mathematical models into executable code.

$$f_{\mathrm{code}} : \mathcal{M} \to \mathcal{C}, \quad M \mapsto C, \tag{2}$$

where $\mathcal{C}$ denotes the space of executable code, and $C \in \mathcal{C}$ is the code generated from the model $M$.

**Debugging Agent:** A workflow that repairs errors in both code and mathematical models using LLMs.

$$\delta_C : \mathcal{C} \times \mathcal{E} \to \mathcal{C}, \qquad\qquad (C, e) \mapsto C \tag{3}$$

$$\delta_M : \mathcal{M} \times \mathcal{C} \times \mathcal{E} \to \mathcal{C}, \qquad\qquad (M, C, e) \mapsto C \tag{4}$$

$$\mathrm{Exec} : \mathcal{C} \to (\mathcal{S} \cup \{\bot\}) \times \mathcal{E}, \qquad\qquad C \mapsto (S, e) \tag{5}$$

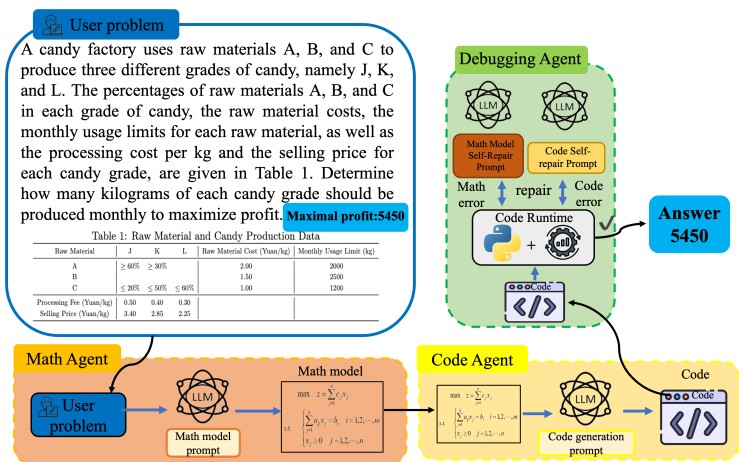

Figure 2: Workflow of OR-LLM-Agent framework.

Here, $\delta_C$ denotes the code repair operator, while $\delta_M$ denotes the mathematical model repair operator. $\mathcal{E}$ represents the space of error messages, with $e \in \mathcal{E}$ being a specific error message. $\mathrm{Exec}$ is the execution operator, and $\mathcal{S}$ denotes the solution space. The symbol $\perp$ indicates a failure to solve the problem or the absence of a feasible solution. The LLMs used in these sub-agents can be any reasoning LLMs, such as GPT-o3, Gemini 2.5 Pro, or DeepSeek-R1.

## 4.2 MATH AGENT AND CODE AGENT

The Math Agent is responsible for generating the mathematical formulation of the problem based on its natural language description. As illustrated in Figure 3, the original problem is presented in LaTeX format. Guided by the system prompt, the LLM constructs the mathematical model and presents it in Markdown format. The Code Agent focuses on generating solver code, as shown in Figure 4. The mathematical model generated by the Math Agent is included in the prompt context, and the LLM is prompted to generate Python code compatible with the Gurobi solver Gurobi Optimization, LLC (2023).

Given that reasoning LLMs do not require overly complex instructions, we provide clear and concise descriptions of the key requirements for each task in prompts. Moreover, by breaking the problem down into well-structured steps, the LLM can effectively leverage its strengths within each specific sub-task, enabling accurate completion of both modeling and coding tasks.

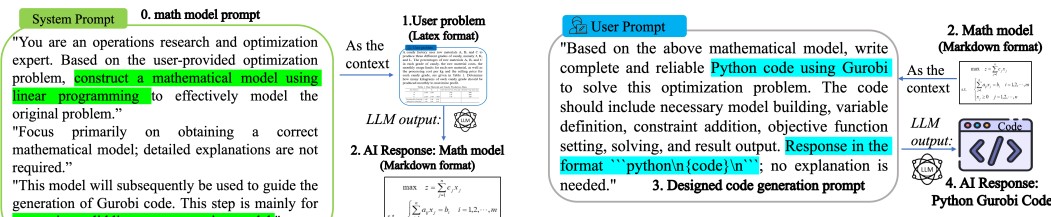

Figure 3: Math model prompt in Math Agent.

Figure 4: Code generation prompt in Code Agent.

## 4.3 DEBUGGING AGENT

The Debugging Agent is responsible for debugging, repairing, and executing the generated code. Its workflow is shown in Figure 5 and Algorithm 1. When code generated by the Code Agent is submitted to the Debugging Agent, it is executed in a Python environment with Gurobi installed. If the execution succeeds and returns an answer, the process terminates and outputs the result. Other-

wise, the sub-agent enters a self-repair phase based on the number of failed attempts. If the number of attempts is fewer than 4, the sub-agent triggers a Code Self-repair process. If the fourth attempt fails, it initiates a Math Model Self-repair. After 5 unsuccessful attempts, the process is terminated and marked as failed. The Code Self-repair is designed to fix runtime code errors. Its workflow is shown in Figure 6. When an error occurs, the error information is appended to the context, and the LLM is prompted to regenerate the corrected code. The Math Model Self-repair is invoked when repeated Code Self-repair fixes fail. In this case, the sub-agent assumes that the error may originate from the underlying mathematical model. The LLM is prompted to re-examine the mathematical model and generate new code accordingly. This workflow is illustrated in Figure 7.

---

**Algorithm 1** OR-LLM-Agent Procedure

1: $M \leftarrow f_{\text{math}}(T)$
2: $C_0 \leftarrow f_{\text{code}}(M)$
3: **for** $attempt = 1, 2, 3, 4, 5$ **do**
4:     $(S_i, e_i) \leftarrow \text{Exec}(C_i)$
5:     **if** $S_i \neq \perp$ **and** $e_i$ indicates success **then**
6:         **return** $S_i$
7:     **end if**
8:     **if** $attempt = 5$ **then**
9:         **return** failure to solve $\perp$
10:     **else if** $attempt = 4$ **then**
11:         $C_{i+1} \leftarrow \delta_M(M, C_i, e_i)$
12:     **else**
13:         $C_{i+1} \leftarrow \delta_C(C_i, e_i)$
14:     **end if**
15: **end for**

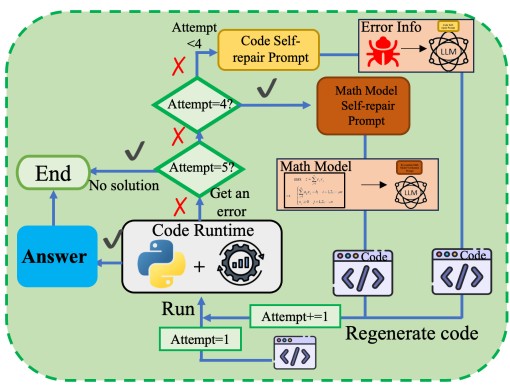

Figure 5: Debugging Agent workflow.

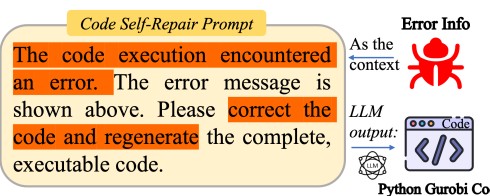

Figure 6: Code self-repair process.

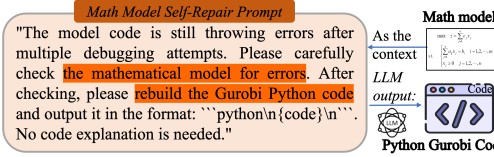

Figure 7: Math model self-repair process.

## 5 BWOR: A BENCHMARK FOR LLMS ON OR TASKS

We release BWOR, an OR benchmark dataset consisting of 82 problems collected from standard OR textbooks (Hu, 2010; 2012). Each problem is presented in LaTeX-formatted natural language, with tabular data included where applicable. These problems are grounded in real-world OR scenarios and require mathematical modeling and solver-based optimization to obtain optimal solutions.

Ground-truth answers are partially sourced from the textbooks and partially computed by domain experts. Since the original problems were written in Chinese, all content was manually translated into English to ensure clarity and consistency. An overview of the dataset is provided in Figure 8.

## 6 EXPERIMENTAL EVALUATIONS

### 6.1 EXPERIMENTAL SETUP

**Benchmarks and metrics**: We use NL4OPT (Ramamonjison et al., 2022), MAMO (Huang et al., 2024), IndustryOR (Huang et al., 2025), and BWOR as evaluation benchmarks. NL4OPT is a widely used benchmark in OR, consisting of 289 linear programming problems. MAMO is a concurrent

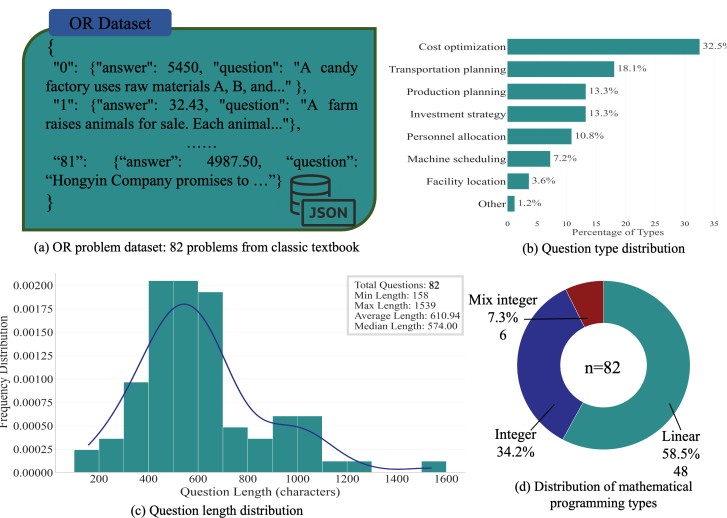

Figure 8: Overview of the BWOR dataset for OR problems.

project designed to assess the mathematical modeling capabilities of LLMs. It includes the EasyLP dataset with 652 simple linear programming problems and the ComplexLP dataset with 211 challenging problems. IndustryOR is an industrial-scale benchmark proposed in ORLM, containing 100 real-world OR problems collected from 8 different industries. The NL4OPT, MAMO, and IndustryOR datasets used in the experiments are obtained from the ORLM repository (Huang et al., 2025). BWOR is an OR dataset, consisting of 82 OR modeling and solving problems. We use accuracy as the evaluation metric, where a prediction is deemed correct if its absolute error from the ground truth is below 0.1.

**Baselines**: To ensure comprehensive evaluation, we compare OR-LLM-Agent against different methods, including SOTA methods, advanced reasoning LLMs, and open-source LLMs. The SOTA baselines include tag-BART (Kani & Gangwar, 2022), Chain-of-Experts (Xiao et al., 2023), OptiMUS (AhmadiTeshnizi et al., 2024), and ORLM (Huang et al., 2025). The advanced reasoning LLMs include GPT-o3, GPT-o4-mini, Gemini 2.5 Pro, and DeepSeek-R1 (Guo et al., 2025). The non-reasoning LLMs include GPT-4o, Gemini 2.0 Flash, and DeepSeek-V3 (Liu et al., 2024). Open-source LLMs include LLAMA3-8B and DeepSeek-R1-Distill-32B. For reasoning and non-reasoning LLMs, we adopt the same prompt template that instructs the LLM to directly generate Python Gurobi (Gurobi Optimization, LLC, 2023) solver code. For open-source LLMs, we use the prompt from ORLM (Huang et al., 2025), which asks the model to generate both the mathematical model and the Python COPT (Ge et al., 2022) solver code.

## 6.2 VALIDATION OF OR DATASET

We evaluate the solution accuracy of various reasoning and non-reasoning LLMs across five OR datasets. All experiments are conducted under a unified setting: each model is prompted to directly generate solver code, which is then executed to obtain the final result. The experimental results are shown in Figure 9.

Notably, on NL4OPT, MAMO-Easy, MAMO-Complex, and IndustryOR, reasoning LLMs from the same series perform worse than their non-reasoning LLMs. For example, on IndustryOR, GPT-o4-mini achieves 5.00% lower accuracy than GPT-4o; on MAMO-Complex, Gemini 2.5 Pro underperforms Gemini 2.0 Flash by 7.11%; on MAMO-Easy, for each model family, the reasoning version yields lower accuracy than its non-reasoning counterpart, with a decline ranging from 1.84% to 17.79%.; and on NL4OPT, Gemini 2.5 Pro performs 4.90% worse than Gemini 2.0 Flash.

These results are unexpected, as reasoning LLMs have consistently demonstrated superior performance over non-reasoning models in mathematics and programming tasks. This suggests they should also exhibit stronger capabilities in solving OR problems. To investigate this hypothesis, we

report benchmark results of LLMs on mathematical reasoning (Balunovic et al., 2025) and coding tasks (Jain et al., 2024), as shown in Figure 10. The results show that reasoning models outperform their non-reasoning counterparts by an average of 63.2% in mathematical reasoning and 47.33% in programming tasks.

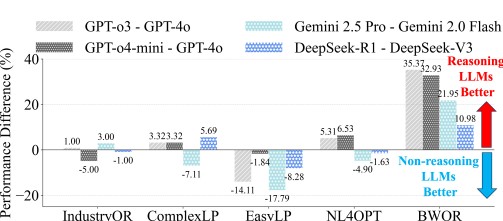

Figure 9: Comparative accuracy of reasoning vs. non-reasoning LLMs on OR datasets.

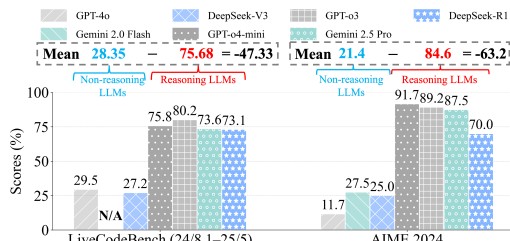

Figure 10: Scores of LLMs on code (Live-CodeBench) and math (AIME 2024) benchmarks. N/A indicates that the model was not included in this benchmark.

In contrast, on BWOR, reasoning LLMs consistently outperform their non-reasoning counterparts from the same model families, with accuracy improvements ranging from 10.98% to 35.37%. This aligns well with their demonstrated strengths in mathematical and programming tasks. It suggests that BWOR provides a more accurate reflection of model capabilities on OR tasks.

In summary, we adopt BWOR as the main benchmark in the following experiments. In other benchmarks, reasoning LLMs often underperform their non-reasoning counterparts within the same model family. Therefore, the results in these datasets are retained for supplementary analysis but are excluded from core comparisons.

## 6.3 EVALUATION ON OR TASKS

To evaluate the performance of LLMs on OR tasks, we conduct experiments across five OR datasets, comparing OR-LLM-Agent with different LLMs as sub-agents, SOTA methods, reasoning LLMs, non-reasoning LLMs, and open-source models. The accuracy results are shown in Table 1.

Based on our earlier analysis of the dataset, we adopt BWOR as the primary benchmark for performance evaluation. Results on the other four datasets are reported as supplementary material for reference by other researchers. On BWOR, OR-LLM-Agent(DeepSeek-R1) achieves the highest accuracy of 82.93%, followed by OR-LLM-Agent(Gemini 2.5 Pro) at 80.49%, and OR-LLM-Agent(GPT-o3) at 79.27%. Among all reasoning models, GPT-o3 performs the best, reaching 75.61% accuracy. For non-reasoning models, the best performer is DeepSeek-V3, with an accuracy of 62.20%. In contrast, ORLM and open-source models yield accuracies below 30%.

These results indicate that OR-LLM-Agent(DeepSeek-R1) demonstrates strong performance on OR tasks, consistently outperforming existing approaches in accuracy.

## 6.4 ABLATION STUDY ON TASK DECOMPOSITION IN OR-LLM-AGENT

To assess whether task decomposition improves the performance of OR-LLM-Agent on OR tasks, we conduct a series of ablation studies on the BWOR dataset. Specifically, we design the following settings:

- Direct Code Generation: The LLM directly generates solver code based on the problem description, without an explicit mathematical modeling process.

- Math Agent + Code Agent: This variant corresponds to OR-LLM-Agent without the Debugging Agent. The LLM first performs mathematical modeling, then generates solver code based on the model, with modeling and coding treated as two separate steps.

Table 1: Comparison of accuracy performance on the NL4OPT, MAMO, IndustryOR, and BWOR benchmarks. Explanation of superscripts: [1] indicates values directly copied from original papers; [2] indicates results reproduced by running the prompt provided in the ORLM GitHub repository; [3] indicates that the result could not be obtained due to LLM service interruption.

| Group | Model (%) | IndustryOR | ComplexLP | EasyLP | NL4OPT | BWOR |
|---|---|---|---|---|---|---|
| *OR-LLM -Agent* | OR-LLM-Agent(GPT-o3) | 34.00 | 51.66 | 80.52 | 75.92 | 79.27 |
| | OR-LLM-Agent(GPT-o4-mini) | 32.00 | 42.18 | 82.21 | 74.29 | 74.39 |
| | OR-LLM-Agent(Gemini 2.5 Pro) | 36.00 | 48.34 | 78.37 | 75.51 | 80.49 |
| | OR-LLM-Agent(DeepSeek-R1) | 36.00 | $-^3$ | $-^3$ | 71.84 | **82.93** |
| *Reasoning LLMs* | GPT-o3 | 34.00 | 37.44 | 69.94 | 74.69 | 75.61 |
| | GPT-o4-mini | 28.00 | 37.44 | 82.21 | 75.92 | 73.17 |
| | Gemini 2.5 Pro | 33.00 | 39.81 | 68.87 | 75.10 | 71.95 |
| | DeepSeek-R1 | 30.00 | 49.29 | 77.15 | 77.96 | 73.17 |
| *Non-reasoning LLMs* | GPT-4o | 33.00 | 34.12 | 84.05 | 69.39 | 40.24 |
| | Gemini 2.0 Flash | 30.00 | 46.92 | 86.66 | 80.00 | 50.00 |
| | DeepSeek-V3 | 31.00 | 43.60 | 85.43 | 79.59 | 62.20 |
| *SOTA methods* | tag-BART | – | – | – | $47.90^1$ | – |
| | Chain-of-Experts | – | – | – | $64.20^1$ | – |
| | OptiMUS | – | – | – | $78.80^1$ | – |
| | ORLM-LLAMA3-8B | 38.00 | 37.40 | 82.30 | 85.70 | 29.27 |
| *Open-source models* | LLAMA3-8B-Base[2] | 0.00 | 0.00 | 0.00 | 0.00 | 0.00 |
| | LLAMA3-8B-Instruct[2] | 0.00 | 0.00 | 0.00 | 0.00 | 0.00 |
| | DeepSeek-R1-Distill-32B[2] | 0.00 | 0.00 | 0.00 | 0.00 | 0.00 |

- Math Agent + Code Agent + Debugging Agent: The full OR-LLM-Agent framework, where the LLM first models the problem, then generates code, and finally performs automatic repair based on execution results.

We evaluate different LLMs under each configuration. The results are shown in Figure 11.

Table 2: Code error rates of LLMs across OR datasets.

| Group | Model (%) | IndustryOR | ComplexLP | EasyLP | NL4OPT | BWOR |
|---|---|---|---|---|---|---|
| *OR-LLM-Agent* | OR-LLM-Agent(GPT-o3) | 1.00 | 0.00 | 0.00 | 0.00 | 1.22 |
| | OR-LLM-Agent(Gemini 2.5 Pro) | 0.00 | 0.00 | 0.00 | 0.00 | 0.00 |
| | OR-LLM-Agent(DeepSeek-R1) | 3.00 | – | – | 0.82 | 0.00 |
| | OR-LLM-Agent(GPT-4o) | 1.00 | 0.00 | 0.15 | 2.45 | 2.44 |
| | OR-LLM-Agent(Gemini 2.0 Flash) | 2.00 | 0.00 | 0.00 | 0.00 | 0.00 |
| | OR-LLM-Agent(DeepSeek-V3) | 0.00 | 0.47 | 0.15 | 0.00 | 0.00 |
| | **Mean** | 0.52 | | | | |
| *Reasoning and non-reasoning LLMs* | GPT-o3 | 4.00 | 4.74 | 1.07 | 0.41 | 3.66 |
| | Gemini 2.5 Pro | 6.00 | 3.32 | 1.07 | 0.41 | 2.44 |
| | DeepSeek-R1 | 13.00 | 3.32 | 1.38 | 1.22 | 4.88 |
| | GPT-4o | 17.00 | 1.90 | 2.30 | 6.53 | 9.76 |
| | Gemini 2.0 Flash | 8.00 | 3.79 | 0.15 | 0.00 | 4.88 |
| | DeepSeek-V3 | 12.00 | 2.37 | 1.07 | 2.86 | 13.41 |
| | **Mean** | 4.56 | | | | |
| **Gap** | *OR-LLM-Agent − Reasoning and non-reasoning LLMs* | -4.04 (0.52-4.56) | | | | |

Compared to Direct Code Generation, the Math Agent + Code Agent setting improves average accuracy by 4.06%, from 62.20% to 66.26%. This demonstrates that performing mathematical modeling before code generation enhances the model's structural understanding of the problem, which contributes to more accurate code output.

Building on this, adding the Debugging Agent to form the full OR-LLM-Agent framework (Math Agent + Code Agent + Debugging Agent) further improves accuracy by 5.49%, from 66.26% to

71.75%. This indicates that the Debugging Agent effectively identifies and corrects code errors, enabling more problems to be solved successfully and enhancing the model's robustness and performance.

In summary, decomposing the OR task into dedicated stages such as modeling, coding, and debugging enables more focused reasoning at each step, with each sub-agent contributing to a cumulative improvement in performance.

## 6.5 ERROR ANALYSIS

To understand the nature of errors in LLM-based OR problem solving, we evaluate two metrics: the code error rate, defined as the proportion of generated code that fails to execute, and the mathematical model accuracy within runnable code, which reflects the proportion of correctly solved instances among those that run successfully.

On NL4OPT, MAMO-Easy, MAMO-Complex, and IndustryOR datasets, we report code error rates. On BWOR, both code and mathematical model accuracy are evaluated. The results are shown in Table 2 and Table 3. We evaluate two groups: (a) OR-LLM-Agent with different LLMs as sub-agents, and (b) reasoning and non-reasoning LLMs.

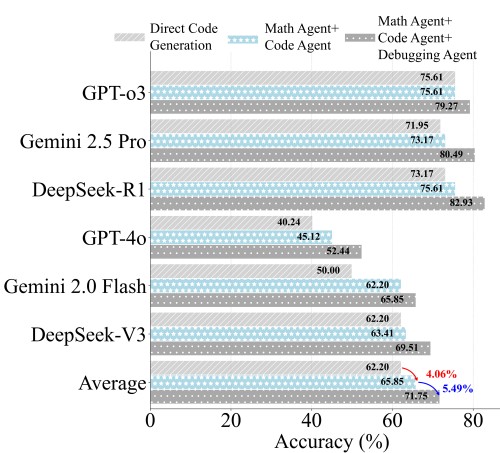

Figure 11: Accuracy across sub-agent configurations.

Table 3: Mathematical model accuracy of LLMs.

| Group | Model (%) | ACC |
|---|---|---|
| OR-LLM-Agent | OR-LLM-Agent(GPT-o3) | 80.25 |
| | OR-LLM-Agent(Gemini 2.5 Pro) | 80.49 |
| | OR-LLM-Agent(DeepSeek-R1) | **82.93** |
| | OR-LLM-Agent(GPT-4o) | 53.75 |
| | OR-LLM-Agent(Gemini 2.0 Flash) | 65.85 |
| | OR-LLM-Agent(DeepSeek-V3) | 69.51 |
| | **Mean** | 72.13 |
| Reasoning and non-reasoning LLMs | GPT-o3 | 78.48 |
| | Gemini 2.5 Pro | 73.75 |
| | DeepSeek-R1 | 76.92 |
| | GPT-4o | 44.59 |
| | Gemini 2.0 Flash | 52.56 |
| | DeepSeek-V3 | 71.83 |
| | **Mean** | 66.35 |
| **Gap** | *OR-LLM-Agent − Reasoning and non-reasoning LLMs* (72.13-66.25) | 5.78 |

Compared to reasoning and non-reasoning LLMs, OR-LLM-Agent reduces the mean code error rate by 4.04%, dropping from 4.56% to 0.52%. On BWOR, it improves mean mathematical model accuracy by 5.78%, from 66.35% to 72.13%. These results suggest that using the OR-LLM-Agent framework enhances code executability and, in most cases, improves mathematical modeling accuracy.

## 7 CONCLUSION

This paper proposes OR-LLM-Agent, an AI agent framework for solving OR problems, built upon reasoning LLMs. The proposed approach enables a fully automated problem-solving pipeline without requiring any additional fine-tuning or complex prompt engineering. OR-LLM-Agent emulates the human problem-solving process by decomposing OR tasks, enabling targeted reasoning and improving overall performance. To evaluate LLM performance on OR tasks, we construct BWOR, an OR benchmark dataset. Compared to existing datasets, BWOR more effectively differentiates models of varying capabilities. Experimental results show the OR-LLM-Agent framework significantly enhances the performance of LLMs on OR tasks while also reducing error rates during problem solving. These results demonstrate the effectiveness of combining reasoning LLMs with task decomposition for solving OR problems.

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
