# OpenReview forum: "OR-LLM-Agent: Automating Modeling and Solving of Operations Research Optimization Problems with Reasoning LLM"
_ICLR.cc/2026/Conference — Submitted to ICLR 2026_

### Official Review · Reviewer_aMoU · 2025-10-29

**Soundness:** 2
**Presentation:** 2
**Contribution:** 2
**Rating:** 2
**Confidence:** 4

**Summary:**

The authors propose a new reasoning LLM agent framework for solving operations research (OR) problems automatically.  The core idea in their framework is to allocate the tasks of modeling, coding, and debugging to three different agents. The authors have also compiled a set of OR problems for which reasoning LLMs outperform non-reasoning LLMs. Then, they compare their agentic framework against different reasoning and non-reasoning LLMs on a set of problems from the literature as well as on their set of problem. Their results demonstrate that decomposing the tasks with their approach yields better accuracy.

**Strengths:**

- There is certainly a growing interest in using LLMs for solving problems requiring complex mathematical modeling.
- The authors' idea of splitting the tasks among three agents does mimick the practice where multiple human agents are involved.
- Unlike the common observation that the reasoning LLMs perform better than their non-reasoning counterparts on mathematical and coding tasks, the authors demonstrate that the non-reasoning LLMs outperform reasoning LLMs on recent datasets published in the literature. This is an interesting finding.
- The comparison and results show that splitting tasks and using reasoning LLM agents can solve OR problems efficiently.

**Weaknesses:**

- The novelty of the paper is limited as it stops with engineering yet another pipeline to solve OR problems with LLMs. "How" and "why" questions are not properly answered.
- The results are demonstrated only on a set of mixed-integer linear problems, of which ~60% is linear programming problems. There are no nonlinear programming or even cone programming problems.
- The third agent is solely responsible for checking the code. However, it is not clear how to automatically control the logical flaws (like negative transshipments), infeasibilities (like insufficient capacity), problem description errors. Since BWOR problems are also compiled from standard OR textbooks, they are already in stylized and clean forms.
- There is no discussion whether the obtained models and codes are the most efficient ones. I believe the authors only evaluate whether the pipeline obtains the optimal solution. In practice, there are many modeling tricks (generating cutting planes, rewriting nonlinear terms, and so on) which may require problem-specific expertise. These tricks could significantly lower the computational effort and time.
- SOTA methods are not fully tested on BWOR models in Table 1.

**Questions:**

- Almost every other day, a new reasoning LLM comes out (e.g., GPT 5). Could we confidently claim that these new models will close the gap on solving BWOR problems when compared against OR-LLM-Agent?
- How can we address optimization problems that are notsuited for mixed-integer models?
- Would the workflow detect the infeasibility in the model or would it consider it as a coding error?
- Why the methods tag-BART, Chain-of-Experts, and ORLM are considered as SOTA approaches?
- What is the impact of the prompt template used to instruct the reasoning and non-reasoning LLMs? Could it be biased toward the pipeline proposed by the authors?
- Why reasoning LLMs consistently outperform their non-reasoning counterparts on BWOR problems? How do these problems differ than the other problem sets proposed in the literature?
- Why are there no results for BWOR problems with three SOTA approaches in Table 1?
- Is there an explanation why none of the open-source models manage to solve a single problem (see last three lines of Table 1)?

---

### Official Review · Reviewer_SPcs · 2025-10-30

**Soundness:** 2
**Presentation:** 2
**Contribution:** 2
**Rating:** 2
**Confidence:** 4

**Summary:**

This paper proposes a novel OR-LLM Agent framework for automated OR problem solving. The core of the framework is to imitate the workflow of humans. It decomposes the task into three sequential stages. Moreover, it constructs BWOR, an OR dataset for evaluating LLM framework performance. Experiments suggest that the OR-LLM-Agent utilizing DeepSeek-R1 outperforms advanced methods.

**Strengths:**

1. The proposed framework is clear and conforms to human thought intuition.
2. This paper constructs and releases a dataset, named BWOR, which can promote the development of OR community.

**Weaknesses:**

1. The claim that the existing methods are constrained by the limited capabilities of non-reasoning LLMs lacks evidence. That means the experiments should include the comparison of the proposed framework underlying reasoning LLMs and non-reasoning LLMs. Moreover, which LLMs are used in the SOTA methods is not clear: reasoning or non-reasoning?
2. The contribution in terms of novelty appears limited. The overall workflow of OR-LLM-Agent lacks originality; compared with CoE, it is more simplified, consisting of only three agents, and its debugging agent closely resembles the evaluator component in CoE.
3. The central focus of the paper is ambiguous — is it the introduction of a new framework or the incorporation of reasoning LLMs? While substituting an LLM is relatively straightforward, proposing a genuinely new framework is considerably more challenging.
4. The paper’s overall clarity needs improvement, particularly in defining what reasoning/non-reasoning LLMs are and in the discussion of existing SOTA methods.
5. The prompts used in the experiments are not provided.

**Questions:**

1. Could you clearly clarify the key distinction between OR-LLM-Agent and existing OR studies that also utilize reasoning-capable LLMs?
2. In Figure 9, reasoning LLMs appear to underperform compared to non-reasoning models. Have you examined alternative factors—such as prompt design or problem characteristics—that might explain this outcome?
3. How fair is it to compare OR-LLM-Agent, as a multi-agent framework, with individual reasoning and non-reasoning models?

---

### Official Review · Reviewer_EPyr · 2025-10-31

**Soundness:** 2
**Presentation:** 3
**Contribution:** 2
**Rating:** 2
**Confidence:** 5

**Summary:**

This paper introduces OR-LLM-Agent, a framework for solving Operations Research (OR) problems using "Reasoning LLMs" without fine-tuning. The agent employs a three-stage task decomposition (mathematical modeling, code generation, and debugging). The authors also propose BWOR, a new benchmark of 82 OR problems manually collected from textbooks. Experiments show the OR-LLM-Agent, when paired with a strong base LLM, outperforms baseline methods on BWOR and other existing benchmarks.

**Strengths:**

1. **Practical Problem and Clear Framework:** The paper addresses the significant and practical challenge of making complex OR problem-solving accessible to non-experts. The proposed OR-LLM-Agent framework is clear and logically structured, following a human-like process of modeling, coding, and debugging.
2. **Contribution of a New Benchmark:** The authors have put forth a non-trivial effort to create and release BWOR, a new dataset for OR problem-solving.
3. **Comprehensive Experimental Validation:** The paper provides solid empirical validation through both ablation studies and error analysis , demonstrating that task decomposition improves not only solution accuracy but also reduces code errors and enhances modeling correctness.

**Weaknesses:**

1. **Limited Novelty in Agent Architecture:** The proposed 3-stage pipeline (model, code, debug) is a very standard and straightforward task decomposition. The paper positions itself against more complex agent systems like Chain-of-Experts (CoE) but does not clearly articulate the architectural novelty or advantage of this design. The primary driver of the improved performance seems to be the use of a newer, more powerful base LLM (e.g., DeepSeek-R1) rather than a novel agent framework.


2. **Unfair Experimental Comparison to SOTA:** The main performance comparison in Table 1 is fundamentally flawed. The paper compares OR-LLM-Agent(DeepSeek-R1) against prior SOTA methods like Chain-of-Experts(GPT-3.5-turbo) and OptiMUS(GPT-4). This comparison confounds the contribution of the agent framework with the contribution of the underlying base LLM. A superior base model will almost always lead to superior results, regardless of the agent framework. The claim that this agent framework is superior to SOTA is therefore unsupported by the provided evidence.

3. **Flawed Justification and Missing Rationale for the BWOR Dataset:** The methodology for introducing the BWOR dataset is scientifically unsound. A standard approach would be to first identify specific limitations in existing datasets (e.g., NL4OPT, MAMO), then propose a new benchmark with a clear selection methodology and design rationale to address those limitations. The paper fails to do this. Instead, it inverts the logic: it presents performance results on BWOR (Section 6.2) and uses these results (i.e., that BWOR better highlights the advantage of reasoning LLMs) as a post-hoc justification for the dataset's quality. This is a form of circular reasoning, as the paper lacks any a priori argument for why BWOR is a valid or necessary benchmark based on its design, screening criteria, or intrinsic properties. Moreover, the underlying assumption that reasoning LLMs should inherently outperform non-reasoning LLMs on OR tasks is itself questionable and unvalidated.

**Questions:**

1. The central claim of your paper rests on the comparison in Table 1, which is confounded by the choice of base LLM. To fairly isolate the contribution of your agent framework, could you provide results for the SOTA baselines (e.g., Chain-of-Experts, Optimus) running on the exact same base model you used (e.g., DeepSeek-R1)? Without this, it is impossible to distinguish agent-level improvements from base-model-level improvements.

2. Following up on the previous question, what was the specific motivation or original purpose for constructing this dataset? What were the exact screening criteria used to select these 82 problems from the textbooks?

---

### Official Review · Reviewer_JrWA · 2025-11-11

**Soundness:** 3
**Presentation:** 3
**Contribution:** 2
**Rating:** 2
**Confidence:** 4

**Summary:**

This paper introduces **OR-LLM-Agent**, a multi-agent framework that automates Operations Research (OR) problem solving using _reasoning LLMs_. The framework decomposes the overall task into three specialized sub-agents for _mathematical modeling_, _code generation_, and _debugging_ respectively. The authors propose **BWOR**, a benchmark of 82 textbook-derived OR problems designed to evaluate LLM reasoning and solver accuracy. Experiments including benchmark comparisons, ablation studies, and error analyses, show that task decomposition and automated debugging can improve the model’s accuracy and robustness.

**Strengths:**

This paper provide an insightful analysis and ablation study convincingly demonstrates the contribution of each sub-agent, with quantitative improvements at every stage (modeling → coding → debugging). Error analysis shows substantial reductions in code failure and improved mathematical model accuracy.

**Weaknesses:**

### 1. Novelty in the agent framework

The paper lacks a strong justification for the conceptual or engineering novelty of proposed multi-agent structure lacks strong . Prior works such as **Chain-of-Experts** and **OptiMUS** have already adopted multi-agent frameworks that decompose OR tasks into similar stages of modeling, coding, and debugging.

### 2. Unconvincing motivation for BWOR

The authors claim that

> reasoning models consistently demonstrated superior performance over non-reasoning models in mathematics and programming tasks

and this served as the main rationale to propose BWOR and adopt it as the main benchmark dataset in the experiments. However, this claim is supported only by limited evidence with one math benchmark and one coding benchmark cited. Moreover, there is evidence that reasoning models can perform worse than base models on complex reasoning tasks (e.g., text-to-SQL), where GPT-4o has been shown to outperform o1.

### 3. Insufficient or unsatisfactory experiments
As an agent-based approach, the paper does not compare against baseline agents on the proposed BWOR dataset, which weakens the empirical support for the claimed improvements. On other benchmarks, the proposed OR-LLM-AGENT underperforms compared to existing agent-based methods. Moreover, the ablation study and error analysis provide limited discussion or comparison to baseline agents

**Questions:**

1. In the proposed agent framework, math model correction is only triggered by code compile errors. How does the framework handle cases where the generated model is incorrect but does not produce any runtime errors, i.e., modeling errors that cannot be detected through execution feedback?

2. Please justify the novelty of the proposed agent framework compared with Chain-of-Experts and OptiMUS.

---

### Meta-Review · Area_Chair_K6d3 · 2026-01-06

**Summary:**

The reviewers identified several critical weaknesses that informed the negative decision:

- Limited Technical Novelty: All reviewers noted that the three-stage pipeline (modeling, code generation, and debugging) is a standard approach already utilized by existing frameworks like Chain-of-Experts and OptiMUS. The contribution was viewed as "engineering yet another pipeline" rather than a conceptual breakthrough.

- Confounded Experimental Comparison: A primary reason for rejection was the unfair comparison in the results. The authors compared their agent (using the powerful DeepSeek-R1 reasoning model) against prior SOTA methods that used older models like GPT-3.5-turbo or GPT-4. This makes it impossible to distinguish whether the performance gains come from the agent framework or simply from using a superior base model.

- Circular Reasoning for the BWOR Dataset: Reviewers criticized the motivation for the new BWOR benchmark. The authors appeared to use the fact that reasoning models perform better on BWOR as a post-hoc justification for the dataset's quality, which was described as "scientifically unsound" and a form of circular reasoning.

- Technical Gaps in Debugging: Reviewer pointed out that the "debugging" agent only triggers on code compile errors. It fails to address cases where the mathematical model is logically incorrect but the code still runs (e.g., negative transshipments or modeling tricks), which is a significant limitation in actual operations research practice.

**Reviewer Concerns:**

there is no author rebuttal

**Reviewer Scores:**

I project no score changing due to lack of rebuttal.

---

### Decision · Program_Chairs · 2026-01-26

Reject